# Predictors and Consequences of Cancer and Non-Cancer-Related Pain in Those Diagnosed with Primary and Metastatic Cancers

Kriti Shah [1], David A. Geller [1], Samer Tohme [1], Michael Antoni [2], Cramer J. Kallem [1], Yoram Vodovotz [1], Rekha Ramanathan [1], Raam Naveen [1], MacKenzie Geroni [1], LaNita Devine [1], Aarshati Amin [1], Gauri J. Kiefer [3], Dan P. Zandberg [3], Vincent Reyes [3] and Jennifer L. Steel [1,*]

[1] Department of Surgery, University of Pittsburgh, Pittsburgh, PA 15213, USA; kra01@upmc.edu (K.S.); gellerda@upmc.edu (D.A.G.); tohmest@upmc.edu (S.T.); crk01@upmc.edu (C.J.K.); vodovotzy@upmc.edu (Y.V.); ramanathanr6@upmc.edu (R.R.); ran01@upmc.edu (R.N.); geronimm@upmc.edu (M.G.); devinele@upmc.edu (L.D.); aam01@upmc.edu (A.A.)

[2] Department of Psychology, University of Miami, Coral Gables, FL 33124, USA; mantoni@miami.edu

[3] UPMC Cancer Center, University of Pittsburgh Medical Center, Pittsburgh, PA 15213, USA; kiefergj@upmc.edu (G.J.K.); daz01@upmc.edu (D.P.Z.); reyesv@upmc.edu (V.R.)

* Correspondence: steejl@upmc.edu; Tel.: +1-412-692-2041

**Abstract:** Objectives: The aims of the study were to (1) describe types of pain in cancer patients, (2) examine the predictors and consequences of pain, (3) investigate the association between type of pain and survival, and (4) examine potential biological mediators of pain and survival. Methods: This was a secondary analysis of baseline data from patients diagnosed with cancer. Patients answered questionnaires that assessed sociodemographic characteristics, pain, depression, sleep, and fatigue. Blood was collected and cytokine assays were performed. Analysis of variance, Kaplan–Meier, and Cox regression survival analyses were used to test the aims. Results: Of the 779 patients diagnosed with cancer, the mean age was 63.5 years, 57.8% male, and 90.6% White. Of those who reported pain (total 70.3%), 46.5% stated their pain was cancer-related while 53.5% stated their pain was non-cancer-related. While both cancer and non-cancer-related pain was associated with depressive symptoms, fatigue, and sleep duration, those with cancer-related pain had significantly higher rates of depressive symptoms ($F_{(1,516)} = 21.217$, $p < 0.001$) and fatigue ($F_{(1,516)} = 30.973$, $p < 0.001$) but not poorer sleep ($F_{(1,497)} = 0.597$, $p = 0.440$). After adjusting for sociodemographic, disease-related characteristics, depression, sleep duration, and morphine milligram equivalent, patient reports of cancer-related pain were significantly associated with poorer survival (HR = 0.646, 95% CI = 0.459–0.910, $p = 0.012$) compared to those with non-cancer-related pain, which was not associated with survival (HR = 1.022, 95% CI = 0.737–1.418, $p = 0.896$). Cytokines did not significantly mediate the link between pain and survival. Conclusion: While nearly half of the pain reported was cancer-related, both types of pain resulted in greater symptom burden, but only cancer-related pain was associated with survival.

**Keywords:** cancer; pain; depression; fatigue; sleep; inflammation; cytokines

## 1. Introduction

Palliative care may be defined as enhancing the medical care and quality of life for people with serious, complex, or terminal medical conditions, such as cancer, through targeted symptom management [1]. The American Society of Clinical Oncology recommends that all patients diagnosed with advanced cancer receive palliative care [2]. Pain is one of the most commonly reported symptoms by patients diagnosed with cancer [3,4]. Across all cancer types, the percentage of patients who experience pain is estimated to be around 50%, yet pain is more frequently reported by those who are in advanced stages [3,5,6]. Pain can have a significant impact on quality of life and often interferes with patients' activities of

daily living [4,7]. In fact, patients diagnosed with cancer report higher levels of pain-related interference with daily functioning compared to non-cancer patients with chronic pain [7]. While the rate of pain-related interference is particularly high in this population, not all pain in those diagnosed with cancer is related to cancer itself [8].

Cancer-related pain may be associated with the diagnosis of cancer itself, tests, and/or treatments. Pain associated with the cancer can be chronic (e.g., spinal cord compression, bone pain, neuropathy), acute (e.g., surgical pain), or transient (e.g., chemotherapy side effects). It is critical to understand the type of pain to develop appropriate interventions for the pain targeted (e.g., neuropathy, surgical pain) [9]. Depression, anxiety, and substance use all have been linked to pain intensity and each symptom exacerbates the other [10,11]. Individuals with chronic pain frequently present with comorbid mental health disorders, but this has been studied much less in those diagnosed with cancer and with cancer-related or non-cancer-related pain [12,13].

In the last decade, not only has the type of pain been differentiated in those with cancer (e.g., chronic versus cancer-related) but pain interference, which is not always correlated with pain intensity, has begun to receive greater attention [10,14]. Furthermore, the quality of pain may also be described by patients as evaluative, sensory, or affective descriptors [14]. Affective descriptors of pain such as "stabbing" have been associated with comorbidity of psychiatric symptoms [15]. While it is known that palliative care referral can generally improve pain symptoms due to the sense of security and continuity this setting provides, to our knowledge, no prior studies have gone further, examining the association of the Brief Pain Inventory's descriptive pain quality in patients diagnosed with advanced cancer or in the palliative care setting [16].

Additionally, chronic and cancer-related pain has been linked to survival outcomes, but not consistently, and they have not been studied together in the same cohort of patients [17,18]. Inflammation has been hypothesized to explain the link between pain and poorer survival, but this hypothesis has not been tested in those diagnosed with cancer. The potential that pain in those with cancer is associated with inflammation is significant as the same cytokines have been shown to promote tumor growth and the metastatic spread of cancer and are therefore important in potentially understanding the link between pain and survival [15,16]. Low levels of anti-inflammatory cytokines, such as IL-10 and IL-4, have also been linked to chronic pain in patients but have been studied less in the context of cancer-related pain [19]. Unfortunately, many of the studies investigating the link between pain, inflammation, and survival did not covary for demographic, disease-specific and treatment-specific factors and/or psychiatric symptoms, which are also associated with inflammation and survival [18–23]. Furthermore, these studies do not differentiate cancer-related pain from non-cancer-related pain [18–23].

The objectives of this study were to (1) describe pain quality, intensity, and interference in patients diagnosed with cancer; (2) investigate the sociodemographic predictors and psychological and behavioral consequences of non-cancer and cancer-related pain; (3) examine the association between pain and survival; and (4) investigate circulating cytokines as potential mediators of survival while covarying for sociodemographic, disease-specific, morphine milligram equivalent (MME), and comorbid psychological and behavioral factors associated with inflammation (i.e., depression, fatigue, sleep duration). We hypothesized that patients who report that their pain is due to cancer will have (1) more pain intensity and interference, (2) more predictors and consequences of pain, (3) poorer survival rates, and (4) common underlying cytokines as mediators of survival than those who report their pain is due to other causes.

## 2. Methods

### 2.1. Design and Participants

This cross-sectional study was a secondary analysis of baseline data from prospective studies performed between 2008–2023. The prospective studies were clinical trials (clinicaltrials.gov registration NCT02939755, NCT016450522 and IRB approvals PRO07050143,

PRO12060036, STUDY19050065). Participants, who were referred by their medical team, were enrolled at a tertiary cancer center in the USA and followed for up to seven years, from diagnosis until death or last follow-up. Patients were explained the risks and benefits of the study and given time to consider participation and ask questions. If the patient was interested in participating in the study, written informed consent was obtained. Inclusion criteria for all studies were (1) cancer diagnosis as determined by radiography or biopsy, (2) 21 years of age or older, and (3) fluency in English. Fluency in English was determined by the study coordinator consenting the patient. If the patient was able to understand English without an interpreter during their medical visit as well as was able to repeat back an understanding of the different components of the consent form, they were considered fluent in English. Exclusion criteria included (1) evidence of a thought disorder, hallucinations, delusions, or suicidal ideation. Data were managed using Research Electronic Data Capture (REDCap) software (REDCap, Vanderbilt University, Nashville, TN, USA) [24–26].

*2.2. Instruments*

For the purposes of the study, patients were assessed at least 8 weeks from their last treatment (e.g., surgery, chemotherapy) so as not to assess symptoms like acute pain associated with treatment.

### 2.2.1. Sociodemographic and Disease Information

Sociodemographic data, such as a patient's sex, age, marital status, income, educational level, and race, were collected using a 13-item questionnaire designed specifically for the prospective studies. The 13-item questionnaire also included information such as height, weight, zip code, and employment status. Disease-specific information, such as diagnosis, number of lesions, largest tumor size, and morphine equivalent were obtained from patients' electronic medical records.

### 2.2.2. Pain

Participants filled out questionnaires for the Brief Pain Inventory (BPI), a valid and reliable instrument that measures pain intensity, interference, quality, location, and management with medications [27]. Additionally, participants were classified based on whether they thought their pain was due to (1) cancer or (2) other causes. To assess pain intensity and interference in cancer-related and non-cancer-related pain, the Pain, Enjoyment of Life and General Activity (PEG) scale, a valid and reliable condensed measure of pain derived from the BPI, was adapted and used for analyses [28,29]. Under the quality subsection of the BPI, patients' pain was categorized based on which terms they used to describe the quality of pain (e.g., stabbing, tender). If the number of affective characteristics (e.g., gnawing, exhausting, stabbing, tiring, nagging, miserable, unbearable) reported by the patient were greater than the number of non-affective characteristics (e.g., aching, throbbing, sharp, shooting, tender, burning, penetrating, numb), the individuals' pain was categorized as primarily affective, and if the patient reported a greater number of non-affective descriptors of the pain they were categorized as non-affective [27–29].

### 2.2.3. Depressive Symptoms

The Center for Epidemiologic Studies-Depression (CES-D) assessment is a 20-item questionnaire that involves a 4-point scale, in which participants report the weekly frequency of depressive symptoms ("rarely," "some days," "occasionally," or "most days"); it is known to be reliable and valid in a population of patients with cancer [30,31]. A summed total score of 16 or greater signifies clinical levels of depressive symptoms [30].

### 2.2.4. Fatigue

The FACIT quality of life assessment system includes a 20-item anemia (FACT-An) module that contains a 13-item questionnaire Functional Assessment of Cancer Therapy-Fatigue (FACT-F) subscale [32]. FACT-F scores range from 0 to 52, with higher scores

indicating less fatigue [32]. FACT-F has been shown to be valid and reliable in a range of cancer populations [32].

### 2.2.5. Sleep

The Pittsburgh Sleep Quality Index (PSQI) is a validated and reliable self-reported questionnaire that assesses sleep quality and disturbances in individuals [33]. Specifically, for this study, the question "During the past month, how many hours of actual sleep did you get at night?" was used to quantitatively determine the sleep duration of participants.

### 2.2.6. Cytokines

Serum levels of cytokines IL-10, IL-1β, IL-2, TNF-α, IFN-γ, and IL-1α were measured. IL-10 is anti-inflammatory while IL-1β, TNF-α, IFN-γ, and IL-1α are pro-inflammatory, and IL-2 is a master regulator type cytokine. Blood draws were performed between 8 a.m. and 12 p.m. when possible. To obtain serum, red-top vacutainer tubes were filled with drawn blood without anticoagulant. Serum aliquots were stored in −80 °C freezers. The samples were thawed only once before testing using Luminex™ (Millipore, Billerica, MA, USA). Milliplex Analyst 5.1 software was utilized to calculate standard curve concentrations and minimal detectable concentrations (MDC) for all measured cytokines in pg/mL.

### 2.3. Procedure

The study protocol was first approved by the University of Pittsburgh's Institutional Review Board. Patients were then referred to the project manager by their medical team. Patients who agreed to speak to a study team member were explained the risks and benefits of the study. Written informed consent was obtained from patients prior to the commencement of study activities, which included completing questionnaires or having their blood drawn. Study data were collected and managed using REDCap electronic data capture tools hosted at the University of Pittsburgh [24–26].

### 2.4. Data Analyses

Patient data were entered, verified, and analyzed in IBM SPSS Statistics 25 (IBM Corp., Armonk, NY, USA). Binary variables were coded for analysis, including education level (less than high school, high school, or more), sex (male, female), endorsement of affective pain component (affective pain, non-affective pain), clinical level cutoffs for depression (score < 16, score ≥ 16), and race (White, racial minority), which was included to examine potential racial health inequities. Cytokines were log-transformed for parametric analysis. Descriptive statistics including mean, standard deviation, distribution, and percentages were computed for each sociodemographic variable. Analyses of variance (ANOVA) and chi-square ($\chi^2$) analyses were performed to test differences between patient groups (patients who reported pain and patients who did not report pain). Linear regression analyses were used to determine significant predictors of pain intensity and interference as well as to determine whether pain was associated with serum levels of cytokines after covarying for sociodemographic, disease-specific, and psychosocial factors. Listwise deletion and imputation were used to handle missing data in all models. Kaplan–Meier and Cox regression analyses were performed to test patient survival in months from diagnosis to death or last follow-up.

## 3. Results

In this study, two sets of analyses were performed. The first analysis was performed with patients who reported non-cancer-related pain and the second analysis was performed with patients who specifically reported their pain was due to cancer. These groups will be referred to as "non-cancer-related pain" and "cancer-related pain," respectively.

### 3.1. Patient Characteristics

A total of 779 patients diagnosed with cancer were included in the study. The majority of patients identified as male (57.8%) and White (90.6%). The mean age of participants was 63.5 years (SD = 11.02). Descriptive statistics for other sociodemographic and disease-specific factors are reported in Table 1. A total of 548 (70.3%) participants reported having any pain within the past week, with 255 (46.5%) reporting cancer-related pain and 293 (53.5%) non-cancer-related pain. In univariate analyses, the PEG score was significantly different by diagnosis [$F(3,571) = 8.46$, $p < 0.001$]. Patients with a diagnosis of hepatocellular or cholangiocarcinoma had the highest mean PEG scores 13.81 (SD = 7.81) followed by patients diagnosed with neuroendocrine carcinoma with liver metastases (mean = 11.02, SD = 8.80); patients other primaries and liver metastases (mean = 10.38, SD = 7.68); and finally, patients with gallbladder, pancreatic, stomach, and appendiceal cancers (mean = 10.31, SD = 8.01).

**Table 1.** Sociodemographic and disease-specific characteristics of sample.

| | Total Sample (*n* = 779) | Patients without Pain (*n* = 231) | Patients with Non-cancer-related Pain (*n* = 293) | Patients with Cancer-related Pain (*n* = 255) |
|---|---|---|---|---|
| **Age (Mean, SD)** | 63.5 (11.02) | 64.6 (11.1) | 63.0 (10.96) | 62.01 (10.35) |
| **Gender (*n*, %)** | | | | |
| Male | 450 (57.8) | 149 (58.4) | 301 (57.4) | 149 (58.4) |
| Female | 329 (42.2) | 106 (41.6) | 223 (42.6) | 106 (41.6) |
| **Marital Status (*n*, %)** | | | | |
| Never Married | 63 (8.1) | 19 (7.5) | 44 (8.5) | 27 (10.7) |
| Married or Cohabitating | 517 (66.6) | 185 (72.5) | 331 (63.7) | 154 (61.1) |
| Widowed, Separated, or Divorced | 188 (24.3) | 49 (19.2) | 139 (26.7) | 66 (26.2) |
| Other | 8 (1.0) | 2 (0.8) | 6 (1.2) | 5 (2.0) |
| **Race (*n*, %)** | | | | |
| White | 701 (90.6) | 239 (93.7) | 462 (89.0) | 221 (88.0) |
| Minority | 73 (9.4) | 16 (6.3) | 57 (11.0) | 30 (12.0) |
| **Education (*n*, %)** | | | | |
| High School or Less | 357 (47.0) | 99 (39.6) | 258 (50.6) | 136 (55.1) |
| More than High School | 404 (53.0) | 151 (60.4) | 252 (49.4) | 111 (44.9) |
| **Income meets basic needs (*n*, %)** | | | | |
| Yes | 633 (83.0) | 226 (89.7) | 407 (79.6) | 85 (73.9) |
| No | 140 (17.0) | 26 (10.3) | 104 (20.4) | 30 (26.1) |
| **Diagnosis (*n*, %)** | | | | |
| Gallbladder/Pancreatic/GIST/Duodenal Stomach Cancer | 60 (7.7) | 22 (8.6) | 38 (7.3) | 21 (8.2) |
| Hepatocellular Carcinoma or Cholangiocarcinoma | 342 (43.9) | 88 (34.5) | 254 (48.5) | 123 (48.2) |
| Primary Cancers with Metastases | 299 (38.4) | 122 (47.8) | 177 (33.8) | 85 (33.3) |
| Neuroendocrine Carcinoma | 78 (10.0) | 23 (9.0) | 55 (10.5) | 26 (10.2) |
| **Number of Lesions (*n*, %)** | | | | |
| None | 147 (20.3) | 51 (21.3) | 96 (19.9) | 41 (17.3) |
| 1–2 | 316 (43.7) | 114 (47.5) | 202 (41.8) | 104 (43.9) |
| 3–4 | 107 (14.8) | 30 (12.5) | 77 (15.9) | 34 (14.3) |
| Five or more | 153 (21.2) | 45 (18.8) | 108 (22.4) | 58 (24.5) |

**Table 1.** *Cont.*

| | Total Sample (*n* = 779) | Patients without Pain (*n* = 231) | Patients with Non-cancer-related Pain (*n* = 293) | Patients with Cancer-related Pain (*n* = 255) |
|---|---|---|---|---|
| Largest Tumor Size in cm (Mean, SD) | 3.62 (3.50) | 3.33 (3.54) | 3.77 (3.48) | 4.33 (3.83) |
| Morphine Equivalent (Mean, SD) in MME/day | 47.83 (193.26) | 3.63 (27.01) | 69.94 (232.91) | 75.66 (250.99) |
| Median Survival and 95% CI (months) | 41 (33.25–48.75) | 56 (35.41–76.59) | 48 (33.17–62.83) | 26 (20.02–1.98) |

*3.2. Description of Pain versus No Pain in Patients Diagnosed with Cancer*

There was no difference between patients who reported pain in the last week versus those who did not report pain with regard to age ($F_{(1,722)}$ = 3.401, $p$ = 0.07), gender ($\chi^2$ = 0.07, $p$ = 0.79), marital status ($\chi^2$ = 6.458, $p$ = 0.09), number of lesions ($\chi^2$ = 3.603, $p$ = 0.31), and largest tumor size ($F_{(1,650)}$ = 2.305, $p$ = 0.13). However, there were significant differences between patients who reported pain versus patients who did not with regard to race as those from a racially minoritized group reported pain in the past week (78.1%) compared to White patients (65.9%) ($\chi^2$ = 4.437, $p$ = 0.04). A greater proportion of patients who did not complete high school reported pain (72.3%) compared to those who completed high school (62.5%) ($\chi^2$ = 8.132, $p$ = 0.04). A greater proportion of patients whose household income did not meet their basic needs reported pain (80.0%) compared to patients whose household income did meet their basic needs (64.3%; $\chi^2$ = 12.023, $p$ = 0.01).

Patients who described the quality of their pain in affective terms, compared to those patients who did not use these terms, reported higher levels of depressive symptoms (mean CES-D = 19.9 [SD = 11.64] versus 15.35 [SD = 9.71]; $F_{(1,453)}$ = 20.926, $p$ < 0.001) and fatigue (mean FACT-Fatigue = 24.36 [SD = 11.01] versus 31.52 [SD = 11.25]; $F_{(1,453)}$ = 46.166, $p$ < 0.001) but no differences in sleep duration (mean hours per night = 8.63 [SD = 2.35] versus 8.53 [SD = 2.31]; $F_{(1,422)}$ = 0.171, $p$ = 0.68); see Figure 1.

*3.3. Cancer and Non-Cancer-Related Pain and Survival*

Using Kaplan–Meier analyses, the median survival for those who reported pain due to cancer was poorer (median = 26 months, 95% CI 20.02–31.98) when compared to patients who reported that their pain was non-cancer-related (median = 48 months, 95% CI =33.17–62.83; log rank = 10.04, $p$ = 0.002). Using Cox regression and after adjusting for sociodemographic and disease-related characteristics, depression, sleep duration, and MME; cancer-related pain remained significantly associated with poorer survival (standardized beta= −0.437, HR = 0.646, 95% CI = 0.459–0.910, $p$ = 0.012). See Table 2 and Figure 2. In contrast, non-cancer-related pain was not significantly associated with survival after adjusting for covariates associated with survival (standardized beta = 0.022, HR = 1.022, 95% CI = 0.737–1.418). The PEG for those who had cancer-related pain was not a significant predictor of survival after adjusting for sociodemographic, disease-specific characteristics, depression, sleep, and MME (standardized beta = 0.16, HR = 1.016, 95% CI = 0.980–1.053, $p$ = 0.395).

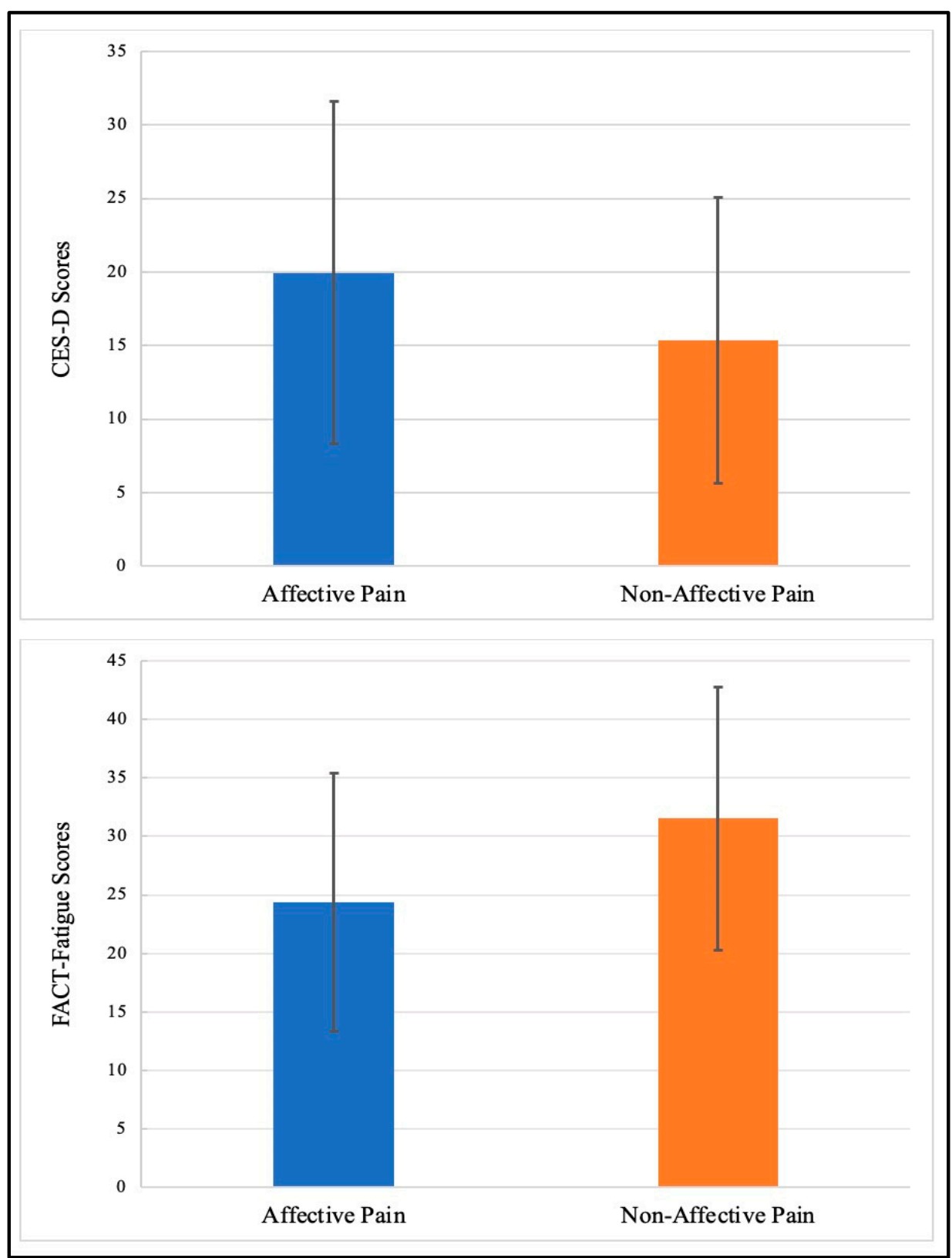

**Figure 1.** Differences in depression (CESD-D) and fatigue (FACT-Fatigue) by descriptors of pain (affective versus non-affective) in patients reporting non-cancer-related pain. A high FACT-Fatigue score reflects less fatigue.

**Table 2.** Cox regression analysis of predictors of cancer-related pain and survival.

| Predictors | B | SE | Sig. | HR | 95.0% CI Lower | 95.0% CI Upper |
|---|---|---|---|---|---|---|
| **Age** | 0.022 | 0.008 | 0.008 | 1.023 | 1.006 | 1.040 |
| **Sex** | −0.236 | 0.190 | 0.215 | 0.790 | 0.544 | 1.147 |
| **Race** | −0.545 | 0.317 | 0.085 | 0.580 | 0.311 | 1.079 |
| **Years of Education** | 0.011 | 0.173 | 0.949 | 1.011 | 0.721 | 1.419 |
| **Diagnosis** | | | 0.125 | | | |
| HCC and CC | −0.086 | 0.400 | 0.830 | 0.918 | 0.419 | 2.011 |
| Other primaries with liver mets | −0.671 | 0.285 | 0.019 | 0.511 | 0.292 | 0.894 |
| Neuroendocrine with liver mets | −0.199 | 0.198 | 0.314 | 0.820 | 0.557 | 1.207 |
| **Number of lesions** | 0.162 | 0.087 | 0.062 | 1.176 | 0.992 | 1.394 |
| **Largest tumor size** | 0.043 | 0.022 | 0.046 | 1.044 | 1.001 | 1.089 |
| **Hours of sleep per night in the past month** | −0.055 | 0.043 | 0.204 | 0.946 | 0.869 | 1.030 |
| **Depressive symptoms** | 0.015 | 0.009 | 0.083 | 1.016 | 0.998 | 1.033 |
| **MME** | 0.000 | 0.000 | 0.748 | 1.000 | 0.999 | 1.001 |
| **Patient-reported cancer-related pain** | −0.437 | 0.175 | 0.012 | 0.646 | 0.459 | 0.910 |

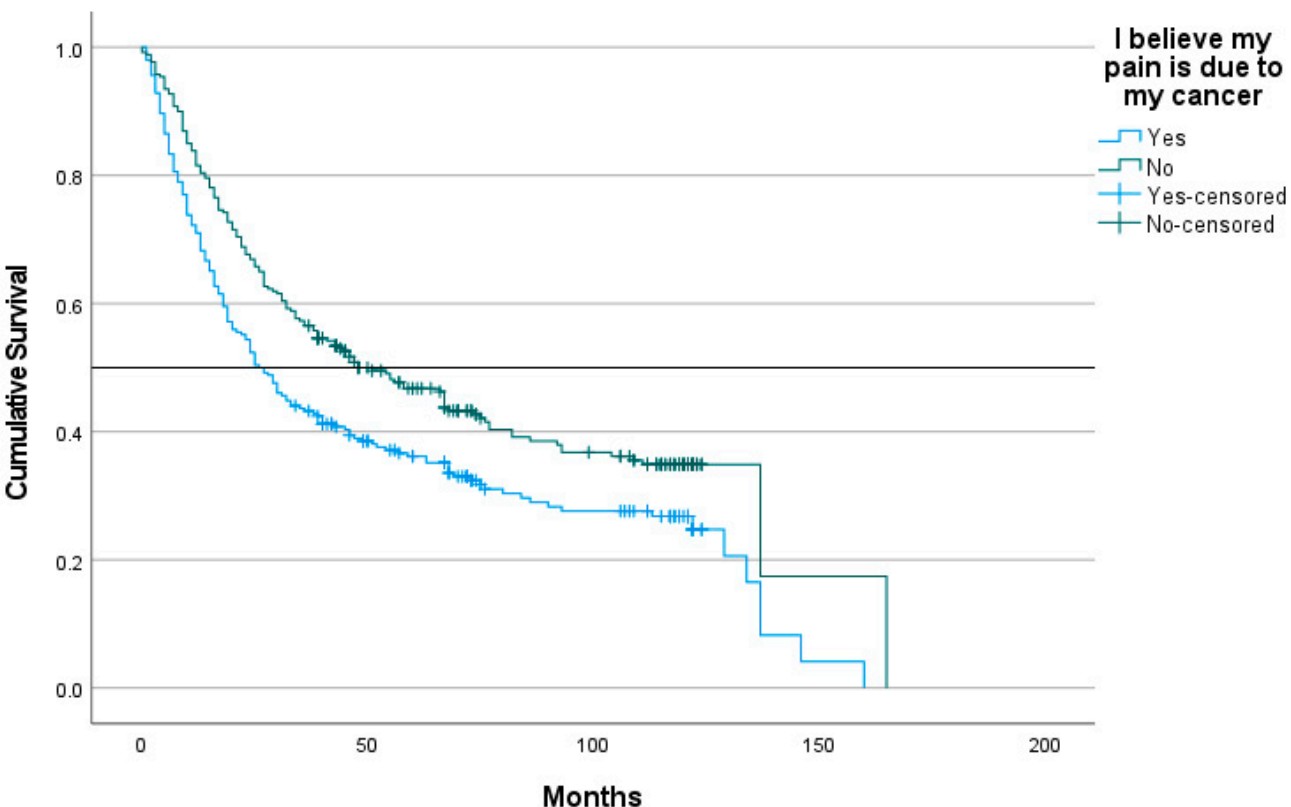

**Figure 2.** Kaplan–Meier survival analysis of patient-reported cancer related pain versus non-cancer-related pain.

### 3.4. The PEG and Circulating Cytokines for Non-Cancer and Cancer-Related Pain

We examined the association between pain and inflammation as a potential mediator between pain and survival. For patients with non-cancer-related pain, after adjusting for sociodemographic, disease-specific, MME, and psychiatric variables as well as patients' daily morphine equivalent, the PEG was not significantly associated with circulating cytokine levels for IL-1β (β = 0.022, 95% CI = −0.031–0.074, $p$ = 0.413), IL-1α (β = 0.022, 95% CI = −0.034–0.078, $p$ = 0.435), TNF-α (β = −0.009, 95% CI = −0.025–0.006, $p$ = 0.234), IFN-γ (β = 0.019, 95% CI = −0.032–0.070, $p$ = 0.463), IL-2 (β = −0.008, 95% CI = −0.030–0.046, $p$ = 0.675), or IL-10 (β = 0.020, 95% CI = −0.022–0.061, $p$ = 0.348).

Similarly, for patients with cancer-related pain, the PEG score was not significantly associated with circulating cytokine levels for IL-1β (β = −0.014, 95% CI = −0.070–0.042, $p$ = 0.627), IL-1α (β = −0.020, 95% CI = −0.074–0.033, $p$ = 0.452), TNF-α (β = 0.008, 95% CI = −0.008–0.023, $p$ = 0.319), IFN-γ (β = −0.029, 95% CI = −0.074–0.017, $p$ = 0.212), IL-2 (β = −0.021, 95% CI = −0.060–0.019, $p$ = 0.300), or IL-10 (β = 0.025, 95% CI = −0.016–0.066, $p$ = 0.222).

### 3.5. Sociodemographic and Disease-Specific Predictors of PEG for Non-Cancer and Cancer-Related Pain

Of the patients who reported non-cancer-related pain, significant predictors of their PEG score included number of years of education (β = −2.80, 95% CI = −4.374, −1.227, $p$ = 0.001) and diagnosis (β = −0.947, 95% CI = −1.551, −0.343, $p$ = 0.001). None of the other sociodemographic or disease-specific variables significantly predicted PEG scores. The predictors included in this model only accounted for 6.4% of the variance in pain scores. When examining cancer-related pain, the only significant predictor of their PEG score was solely the number of years of education (β = −3.975, 95% CI = −6.876, −1.074, $p$ = 0.008). None of the other sociodemographic or disease-specific variables significantly predicted PEG scores (see Table 3). The predictors included in this model only accounted for 8.7% of the variance in pain scores.

**Table 3.** Sociodemographic and disease-specific predictors of cancer-related pain.

| Predictor | Beta | Standard Error | $p$-Value | 95% CI | |
|---|---|---|---|---|---|
| Age | −0.018 | 0.072 | 0.798 | −0.161 | 0.124 |
| Gender | −1.023 | 1.490 | 0.494 | −3.982 | 1.935 |
| Race | 2.614 | 2.261 | 0.250 | −18.74 | 7.102 |
| Education | −3.975 | 1.461 | 0.008 | −6.876 | −1.074 |
| Income | 3.228 | 1.678 | 0.057 | −0.103 | 6.560 |
| Diagnosis | −2.856 | 1.500 | 0.060 | −5.834 | 0.121 |
| Largest Tumor Size | −0.180 | 0.206 | 0.384 | −0.588 | 0.228 |
| Number of Lesions | 0.603 | 0.745 | 0.421 | −0.877 | 2.083 |

### 3.6. The PEG and Psychological and Behavioral Factors

For patients with non-cancer-related pain, after adjusting for significant sociodemographic and disease-specific variables as well as daily morphine equivalent (MME), the PEG predicted significantly higher levels of depressive symptoms (β = 0.587, 95% CI = 0.425, 0.749, $p$ < 0.001), fatigue (β = −0.858, 95% CI = −1.031, −0.684, $p$ < 0.001), and significantly fewer hours of sleep per night (β = −0.055, 95% CI = −0.088, −0.023, $p$ = 0.001). The predictors in the models account for 24.1% of the variance for depression, 30.7% for fatigue, and 6.3% for sleep duration. For patients with cancer-related pain, after adjusting for significant sociodemographic and disease-specific variables as well as daily morphine equivalent, the PEG predicted significantly higher levels of depressive symptoms, higher levels of fatigue, and significantly fewer hours of sleep per night. The predictors in the models account for

16.0% of the variance for depression, 37.8% for fatigue, and 5.5% for sleep duration; see Table 4.

While both cancer and non-cancer-related pain was associated with depressive symptoms, fatigue, and sleep duration, those with cancer-related pain had significantly higher rates of depressive symptoms ($F(1,516) = 21.217$, $p < 0.001$; mean = 18.64, SD = 11.36 versus 14.33, SD = 9.87) and fatigue ($F(1,516) = 30.973$, $p < 0.001$; mean = 26.20, SD = 11.32 versus 31.71, SD = 11.59) but not shorter sleep duration ($F(1,497) = 0.597$, $p = 0.440$).

**Table 4.** Pain intensity and interference (PEG) as a predictor of psychosocial and behavioral factors in patients reporting cancer-related pain.

| Predictor | Beta | Standard Error | *p*-Value | 95% CI | |
|---|---|---|---|---|---|
| **DEPRESSION** | | | | | |
| Age | −0.066 | 0.109 | 0.544 | −0.283 | 0.150 |
| Gender | 2.288 | 2.352 | 0.333 | −2.388 | 6.964 |
| Race | −5.768 | 3.572 | 0.110 | −12.870 | 1.133 |
| Education | −1.526 | 2.364 | 0.520 | −6.225 | 3.173 |
| Income | 2.990 | 2.662 | 0.264 | −2.302 | 8.283 |
| Diagnosis | −0.347 | 0.898 | 0.700 | −2.133 | 1.439 |
| Largest Tumor Size | −0.119 | 0.312 | 0.704 | −2.171 | 2.490 |
| Number of Lesions | 0.159 | 1.172 | 0.892 | −0.739 | 0.501 |
| Morphine Equivalent | 0.007 | 0.004 | 0.108 | −0.002 | 0.015 |
| PEG | 0.541 | 0.154 | 0.001 | 0.235 | 0.848 |
| **FATIGUE** | | | | | |
| Age | −0.136 | 0.094 | 0.149 | −0.322 | 0.050 |
| Gender | −4.820 | 2.016 | 0.019 | −8.829 | −0.811 |
| Race | 4.505 | 3.062 | 0.145 | −1.584 | 10.594 |
| Education | −1.173 | 2.026 | 0.564 | −5.202 | 2.856 |
| Income | −3.674 | 2.282 | 0.111 | −8.212 | 0.864 |
| Diagnosis | 1.798 | 0.770 | 0.022 | 0.267 | 3.330 |
| Largest Tumor Size | −0.327 | 0.267 | 0.224 | −0.858 | 0.204 |
| Number of Lesions | −0.549 | 1.005 | 0.586 | −2.547 | 1.449 |
| Morphine Equivalent | 0.004 | 0.004 | 0.282 | −0.003 | 0.011 |
| PEG | −0.791 | 0.132 | <0.001 | −1.054 | −0.529 |
| **SLEEP DURATION** | | | | | |
| Age | 0.002 | 0.021 | 0.932 | −0.040 | 0.043 |
| Gender | 0.589 | 0.450 | 0.194 | −0.307 | 1.485 |
| Race | 1.024 | 0.675 | 0.133 | −0.319 | 2.367 |
| Education | 0.141 | 0.449 | 0.754 | −0.752 | 1.034 |
| Income | 0.124 | 0.514 | 0.810 | −0.898 | 1.145 |
| Diagnosis | 0.080 | 0.170 | 0.640 | −0.258 | 0.418 |
| Largest Tumor Size | 0.073 | 0.060 | 0.223 | −0.046 | 0.192 |
| Number of Lesions | −0.215 | 0.223 | 0.337 | −0.657 | 0.228 |
| Morphine Equivalent | −0.001 | 0.001 | 0.301 | −0.002 | 0.001 |
| PEG | −0.068 | 0.029 | 0.023 | −0.126 | −0.009 |

## 4. Discussion

Given the severity of a disease like cancer as well as the American Society of Clinical Oncology's recommendations that all patients with advanced cancer receive palliative care, understanding pain in a population of cancer patients is important to providing key symptomatic management [2]. While a large majority of people diagnosed with cancer reported pain, we found only about one-third of patients reported cancer-related pain, which is consistent with prior cancer research [3,34]. Approximately two-thirds of patients with advanced cancer report pain; however, no study has differentiated cancer-related and non-cancer-related pain in advanced cancer or palliative care settings [35]. Cancer-related pain was associated with poorer survival and the association with survival was sustained after adjusting for sociodemographic and disease-specific factors, psychiatric factors (depression, sleep duration), and opioid use, which has previously been associated with poorer survival [35–38]. To our knowledge, this may be the first study to compare cancer-related and non-cancer-related pain with survival in those diagnosed with cancer while adjusting for important covariates associated with survival. Interestingly, it was not the intensity or interference of the pain associated with survival (i.e., as measured with the PEG), but the patients report that the pain was "cancer-related" rather than "non-cancer-related" pain that was associated with survival. These findings have consequences for treatment as well. Since it is known that opioids provide modest pain relief in chronic non-cancer pain and potential for dependence [39], it is especially important to identify whether patients have cancer-related pain rather than non-cancer-related pain, so that chronic opioid treatment is not utilized in cases where it is not warranted.

While some studies have found a link between pain and cytokines, our study did not observe an association between cancer-related or non-cancer-related pain and inflammation as measured by circulating cytokines [36]. The lack of a consistent cytokine association across advanced cancer types may suggest a unique inflammatory process not measured in this study and/or ceiling levels of circulating cytokine levels in this population [37]. As new immunotherapies become more widely prescribed, further understanding of how this treatment may influence circulating cytokines, and the link with pain will be important [38].

Only two sociodemographic and disease-specific variables significantly predicted overall pain intensity and interference: educational attainment and type of cancer diagnosis. Specifically, patients with less than a high school education as well as those diagnosed with hepatobiliary cancers (hepatocellular carcinoma or cholangiocarcinoma) tended to report greater non-cancer-related pain intensity and interference, which is consistent with previous findings [40]. Lower health literacy including the understanding of the causes and treatment options for pain may explain the link between educational level and pain [40–42]. In addition, approximately half of the patients diagnosed with hepatocellular carcinoma may have had chronic exposure to drugs and/or alcohol that may affect their experience of pain, or alcohol and drug use may have been a result of unmanaged chronic physical or emotional pain [43–45]. These predictors accounted for only a small amount of the variance in cancer-related and non-cancer-related pain, suggesting that further research is warranted.

It is known that patients with cancer, as well as the general population, who report clinical levels of depressive symptoms along with those who sleep less than 6 h or more than 10 h per night tend to have poorer survival [46,47]. While both cancer-related and non-cancer-related pain predicted higher levels of depressive symptoms and poorer sleep, those with cancer-related pain had higher levels, possibly further driving the association with poorer survival when compared to those with non-cancer-related pain.

This study has several strengths, including its large sample size and validated measures of pain, fatigue, sleep, and depression as well as the inclusion of several biomarkers of inflammation. To our knowledge, this is the first study to differentiate between non-cancer-related and cancer-related pain in patients diagnosed with cancer while covarying for sociodemographic, psychological, and behavioral factors and MME to examine the relationships between pain, inflammatory biomarkers, and survival in patients diagnosed with cancer. It is also the first study to our knowledge to examine the quality of pain and psychological and behavioral outcomes in patients diagnosed with cancer.

In terms of the limitations of this study, the cohort is predominantly White. Therefore, the experiences of minority patients may not be accurately reflected in this study. Another limitation of this study is that the questionnaires obtained to collect the data were all self-reported by the patients. Furthermore, there is some evidence to suggest that sleep duration and fatigue may be prodromal symptoms of cancer, and this study did not examine the direction of the relationship between symptoms [48]. Future longitudinal studies are needed to determine the chronological order of pain–depression–fatigue–sleep cluster symptoms in cancer patients. Other potential biomarkers may also be explored with regard to the mediation of cancer-related pain and survival [49]. Cognitive-behavioral or new generation cognitive-behavioral interventions (e.g., acceptance commitment therapy) may be able to address nuances in pain appraisals tied to these descriptors to reduce pain and the associated symptoms to improve quality of life in those diagnosed with cancer [50–52]. In the palliative care setting where patients and families often want to limit opioid use to decrease sedation, cognitive-behavioral strategies may be particularly welcomed [53,54]. Along with these recommendations for future research, advancing clinical research in the development of novel and innovative patient-reported outcomes to screen patients and treatments that address both pain and comorbid symptoms in patients diagnosed with cancer is warranted.

**Author Contributions:** Conceptualization: K.S., C.J.K. and J.L.S.; methodology: J.L.S. and C.J.K.; validation: R.N., M.G., L.D. and A.A.; formal analysis: K.S. and J.L.S.; investigation: D.A.G., S.T., Y.V., R.R., G.J.K., D.P.Z. and V.R.; data curation: R.N., M.G. and L.D.; writing—original draft preparation: K.S.; writing—review and editing: D.A.G., S.T., M.A., C.J.K., Y.V., R.N., M.G., L.D. and J.L.S.; visualization: K.S. and J.L.S.; supervision: J.L.S.; funding acquisition: D.A.G., M.A. and J.L.S. All authors have read and agreed to the published version of the manuscript.

**Funding:** This research was funded by the National Cancer Institute (K07CA118576; R21CA127046; R01CA196953; R01CA176809; Clinical and Translation Sciences Institute at the University of Pittsburgh UL1-TR-001857).

**Informed Consent Statement:** Written informed consent was obtained from all subjects involved in the study. All subjects were de-identified in this study.

**Data Availability Statement:** The data presented in this study are available on request from the corresponding author. The data are not publicly available since the prospective trials are currently ongoing.

**Conflicts of Interest:** The authors declare no conflict of interest. Antoni notes that he is a paid consultant for Blue Note Therapeutics and Atlantis Healthcare, both digital health companies.

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
