# Peer review of "Predictors and Consequences of Cancer and Non-Cancer-Related Pain in Those Diagnosed with Primary and Metastatic Cancers"

_curroncol, doi:10.3390/curroncol30100637_

Round 1

Reviewer 1 Report

The authors try to "(1) describe types of pain in cancer patients, (2) examine the predictors and consequences of pain, (3) investigate the association with type of pain and survival, and (4) examine potential biological mediators of pain and survival".

I do not understand the study design, hypothesis, primary outcome. Registration numbers and IRB approval numbers are missing.

Major concerns

#1. Please describe the study design, hypothesis, primary outcome.

#2. Registration numbers and IRB approval numbers are missing.

Minor comments

#1. What is MME (abstract)?

#2. How were the participants enrolled (Page 2, Line 90)?

#3. How did the authors define the “fluency in English” (Page 2, Line 93)?

#4. How did the authors validate the pain status reported by patients (Page 3, Line 105)?

#5. Did the authors measure cytokine levels only once?

#6. I do not understand Tables and Figures. Where are table/figure legends?

None.

Reviewer 2 Report

I would like to thank the editor for the chance to review this original and nicely conducted study.

I have 2 small suggestions:

1.       Especially in light of increasingly prolonged survival of patients with metastatic cancer and the opioid crisis, it is vital that only patients who really benefit from opioids actually receive them. And that does not include those with chronic pain (and, coincidentally, metastatic disease).

In short: opioids for patients with cancer pain and not for patients with chronic pain ánd cancer.

I would appreciate it if the discussion brings this point out a bit more clearly: Can you, in response to the findings in this study, push the discussion a bit more clearly?

2.       The relationship between pain intensity and lower educational level has been found before, including in patients with cancer (e.g. DOI: 10.1016/j.pain.2007.08.022, DOI: 10.1200/JGO.2016.006783). The former also found an association between risk of undertreatment and lower education level.

Round 2

Reviewer 1 Report

The authors improved the manuscript. I still do not understand the figures. Also, the manuscript would be improved by a thorough English language review before acceptance for publication.

Major concerns

#1. What do the authors try to show in Figure 1?

#2. There are only two lines in Figure 2, although variables are four (yes, no, yes censored, and no censored).

#3. The manuscript would be improved by a thorough English language review before acceptance for publication.

None.

Reviewer 2 Report

Thank you for addressing my remarks

Author Response

Thank you!

Round 3

Reviewer 1 Report

The authors have improved the manuscript. I have no further comments.